# A Predictive Model to Analyze the Factors Affecting the Presence of Traumatic Brain Injury in the Elderly Occupants of Motor Vehicle Crashes Based on Korean In-Depth Accident Study (KIDAS) Database

**DOI:** 10.3390/ijerph18083975

**Published:** 2021-04-09

**Authors:** Hee Young Lee, Hyun Youk, Oh Hyun Kim, Chan Young Kang, Joon Seok Kong, Yeon Il Choo, Doo Ruh Choi, Hae Ju Lee, Dong Ku Kang, Kang Hyun Lee

**Affiliations:** 1Automotive Medical Science Research Institute, Wonju College of Medicine, Yonsei University, Wonju 26426, Korea; hylee3971@yonsei.ac.kr (H.Y.L.); yhmentor@gmail.com (H.Y.); ardentem@gmail.com (O.H.K.); chlrhkcy@naver.com (C.Y.K.); jskong4208@gmail.com (J.S.K.); yg1tym@naver.com (Y.I.C.); dooruh78@naver.com (D.R.C.); emt_hju@naver.com (H.J.L.); gound333@nate.com (D.K.K.); 2Department of Emergency Medicine, Wonju College of Medicine, Yonsei University, Wonju 26426, Korea

**Keywords:** traumatic brain injury, the elderly, predictive model, validation analysis, KIDAS database

## Abstract

Traumatic brain injury (TBI), also known as intracranial injury, occurs when an external force injures the brain. This study aimed to analyze the factors affecting the presence of TBI in the elderly occupants of motor vehicle crashes. We defined elderly occupants as those more than 55 years old. Damage to the vehicle was presented using the Collision Deformation Classification (CDC) code by evaluation of photos of the damaged vehicle, and a trauma score was used for evaluation of the severity of the patient’s injury. A logistic regression model was used to identify factors affecting TBI in elderly occupants and a predictive model was constructed. We performed this study retrospectively and gathered all the data under the Korean In-Depth Accident Study (KIDAS) investigation system. Among 3697 patients who visited the emergency room in the regional emergency medical center due to motor vehicle crashes from 2011 to 2018, we analyzed the data of 822 elderly occupants, which were divided into two groups: the TBI patients (N = 357) and the non-TBI patients (N = 465). According to multiple logistic regression analysis, the probabilities of TBI in the elderly caused by rear-end (OR = 1.833) and multiple collisions (OR = 1.897) were higher than in frontal collision. Furthermore, the probability of TBI in the elderly was 1.677 times higher in those with unfastened seatbelts compared to those with fastened seatbelts (OR = 1.677). This study was meaningful in that it incorporated several indicators that affected the occurrence of the TBI in the elderly occupants. In addition, it was performed to determine the probability of TBI according to sex, vehicle type, seating position, seatbelt status, collision type, and crush extent using logistic regression analysis. In order to derive more precise predictive models, it would be needed to analyze more factors for vehicle damage, environment, and occupant injury in future studies.

## 1. Introduction

Drivers must use visual and auditory senses to recognize the surrounding situation and make appropriate judgments and responses based on the perceived situation. Studies show that the elderly drivers may have a decreased ability to cope with other traffic situations due to physical problems (aging or chronic disease) and reduced mental abilities, resulting in motor vehicle crashes (hereinafter, MVCs) [1,2,3]. However, there is still much disagreement about the age standard for elderly drivers. As the world’s population ages, the classification criteria for the age of the elderly changes. According to the United Nations (UN), the life expectancy of the world has increased from 69.07 in 2005–2010 to 70.79 in 2010–2015, and the proportion of the population over 60 years of age rose from 9.9% in 2005 to 12.3% in 2015, which is expected to increase to 16.5% in 2030 and 21.5% in 2050 [4,5].

The percentage of total deaths from MVC victims over the age of 65 in OECD countries increased from 21.7% in 2009 to 24.0% in 2018 [5]. Elderly drivers have a high mileage death rate, and commonly get into MVCs due to speeding and in intersections. A study analyzing the 1993–2004 data from the National Automotive Sampling System-Crashworthiness Data System (NASS-CDS) found that the elderly drivers (over 65 years of age) had a different incidence of severe injuries (MAIS 4+) in other age groups (16–24, 25–44, and 45–64 years old). It also found a 4.3 times greater risk of serious injury than younger drivers (16–24 years old).

Traumatic brain injury (TBI), also known as intracranial injury, occurs when an external force injures the brain. Fernandes and Sousa introduced head injuries and their mechanisms and reviewed, in detail, the thresholds for head injuries and the criteria for each head injury [6]. TBI can be classified based on severity, mechanism, or other features. According to the World Health Organization, TBI will become the third leading cause of death and disability globally by 2020 [7]. The elderly TBI patients will be increasingly burdened by society with an aging population worldwide. Peschman found that age alone increases the likelihood of being hospitalized after a head injury [8].

In Europe, the overall incidence is 262 per 100,000 persons per year for patients with TBI admitted to hospitals, and across developed countries, the annual incidence is somewhat lower at approximately 200 per 100,000 admissions [9,10,11]. However, these incidences may be underestimated since most individuals with mild TBI are not hospitalized [12]. However, in recent studies, it has been emphasized that the health policy for injury prevention should be focused on groups with high risk of injury [13].

In the United States, 5.3 million people have TBI-related disabilities (i.e., 1~2% of the population), and 7.7 million European survivors of brain trauma have disabilities [14]. Deficits associated with TBI include impaired attention, poor executive function, impulsivity, poor decision-making, and aggressive behavior [15]. Such deficits may affect driving performance and pose a risk of MVCs. Unfortunately, a large percentage of TBIs are sustained through MVCs [16]. Many individuals choose not to drive after TBI because of their impairments, and others will no longer be able to drive because of the severity of their injuries. In a cohort study conducted in Europe, it was reported that the possibility of returning to society can be increased if severe damage is quickly predicted and rehabilitation and psychological treatment are carried out quickly for victims of motor vehicle crashes [17].

In this study, although there are many reports that recently raised the age criteria for classification of the elderly, the criterion for being elderly was conservatively set at 55 in order to more seriously consider the factors affecting traumatic brain injury in the elderly. The Centers for Disease Control and Prevention in the United States recommends special consideration for older adults which is not applicable for stages 1 to 3 over 55 years old (increased risk of injury and death) and over 65 years old (suspected to have a shock if the systolic blood pressure is less than 110 mmHg) in the Guidelines for Field Triage of Injured Patients in 2011 [18]. Ashok reported that older drivers aged 55 or older face greater risks on the road through the correlation between break reaction time (BRT) and age [19].

The purpose of this study was to analyze the factors affecting the presence of TBI in the elderly motor vehicle occupants (hereinafter, MVOs) based on the database about the MVCs in Korea and to develop a predictive model to determine the presence of the traumatic brain injuries in the elderly occupants. Although this study was not the first to investigate the risk factor for TBI in MVCs, it was innovative because of two reasons. First, this study was aware of the problem of TBI among the elderly by focusing on the increasing number of the elderly MVOs in an aging society in Korea. Second, this study analyzed and derived a predictive model by using real-world evidence based on an actual investigation system in Korea. In addition, the explanatory power of the model was estimated by presenting the sensitivity, specificity, and accuracy through the external validation analysis.

## 2. Materials and Methods

### 2.1. Data Source

This was a retrospective observational study based on a real-world investigation system in MVCs. Data from the Korean In-Depth Accident Study (KIDAS) database corresponding to January 2011 to December 2018 were collected from on-scene investigations that detailed information concerning MVOs who had visited emergency medical centers in five cities [20]. This study was conducted following approval from the research ethics committee of Wonju Severance Christian Hospital, Yonsei University (IRB Approval No.: CR313137).

### 2.2. Data Acquisition

#### 2.2.1. Crash Data—Collision Deformation Classification Code (CDC Code)

The Collision Deformation Classification (CDC) code is a vehicle damage code devised by the Society of Automotive Engineers (SAE) that indicates vehicle damage areas, damage shapes, and the degree of damage. The seven-character code is also an expression useful to persons engaged in automobile safety to appropriately describe a field-damaged vehicle with conciseness in their oral and written communications. The classification system consists of seven characters—three numeric and four alphameric—arranged in a specific order. The characters describe the deformation detail concerning the direction, location, size of the area, and extent which, combined together, form a descriptive composite of the vehicle damage [21].

#### 2.2.2. Injury Data—Abbreviated Injury Scale (AIS)/Injury Severity Score (ISS)

The Abbreviated Injury Scale (AIS) is an anatomical-based coding system created by the Association for the Advancement of Automotive Medicine to classify and describe the severity of injuries [22]. AIS is already known as a useful anatomical-based coding system for classifying and describing the severity of injuries due to MVCs, for example, as severe patients. Based on analyzing the whole medical record in an electronic clinical information system, two coordinators coded and two observers rechecked by using 05/08 update of AIS manual [23]. The two observers, who included one paramedic and one resident in specialist training, had been trained for this study. For classifying each body region, AIS08 consists of head, face, neck, thorax, abdomen, spine, upper extremity, lower extremity, external, and other. The maximum abbreviated injury scale (MAIS) expresses the maximum AIS value from all body regions. In this study, the case where the TBI occurred through the AIS-head code was assigned was classified as a ‘TBI group’, and the case where the TBI did not occur was classified as a ‘non-TBI group’. Carroll et al., referring to the revised version of AIS-2005, expressed an assessment of the TBI compared to previous versions. We proceeded to classify the TBI through this and classified it into two groups according to the occurrence [24].

Furthermore, the injury severity score (ISS) is an injury scale that rates the severity of injury for each patient, calculated by summing the squares of AIS values with the highest severity scores from 3 body regions [25]. For ISS calculation, AIS codes were regrouped into the following body regions: head and neck area; facial area; chest area; internal organs in the abdominal and hip area; upper and lower extremities and hips; and external factors such as burn, frostbite, and explosion. The ISS is expressed numerically between 0 and 75 points, and a patient with a score of 16 points is deemed to be severe with multiple injuries [26]. Patients with an AIS 6 injury are automatically designated ISS 75.

#### 2.2.3. Definition of the Controlled Indicators

TBI was defined through AIS codes related to the head, but these do not include codes for surface-level injuries like contusions, abrasions, or minor lacerations. MVOs were classified according to their seating position in the vehicle as either the driver, front passenger, second-row left passenger, or second-row right passenger. Vehicle types were classified into sedan, sport utility vehicle (SUV), light truck, or van. The collision types were classified as frontal collision, lateral-nearside collision, lateral-farside collision, or rear-end collision according to Column 3 (General area of damage) of the CDC code, and the ‘O’ cases of Column 6 (General type of damage distribution) of the CDC code were classified as rollover. In addition, cases of two CDC codes or more were defined as multiple collisions. The crush extent in MVCs was expressed by the seventh column of the CDC code. The zones were classified according to the degree of vehicle deformation as Zone 1 (CE Zone 1), Zone 2 (CE Zone 2), or Zone 3 (CE Zone 3). The results in the emergency room were classified into discharge, transfer, ward admission, intensive care unit (ICU) admission, and expired. Similarly, the results in the admission were classified into discharge, transfer, or expired.

#### 2.2.4. Inclusion and Exclusion Criteria

Of a total of 3697 MVOs involved in MVCs between 2011 and 2018, we used data from 822 elderly MVOs. We excluded MVOs by several conditions which were under 55 years old (N = 2118), 6th columns in CDC code ‘Sideswipe’ (N = 115), and incomplete data (N = 576). It was divided into two groups as the TBI patients (N = 357) and the non-TBI patients (N = 465) (Figure 1).

### 2.3. Data Analysis

#### 2.3.1. Logistic Regression Model

The predictive model was developed using data collected from 2011 to 2018 in the KIDAS database. The logistic regression model was expressed through the following equation (Equation (1)).
(1)$$P\left(Y=1\right)=\frac{e^{A}}{1+e^{A}}\left(A:\;\beta 0+\beta 1\times 1+\beta 2\times 2+\ldots +\beta n\times n\right)$$
where *P*(*Y* = 1) is the probability of an occurrence of TBI in the elderly MVOs and A is the partial regression coefficient of each x, estimated using the maximum-likelihood method. The x-values represent the factors of TBI in the elderly MVOs, including vehicle type, seatbelt status, seat position, seat row, collision type, collision range according to CDC code, sex, and age. The reference for each variable was set as follows: Male, the vehicle type was set to Sedan, the sitting position was set to Driver, the seatbelt status was set to Fastened, the collision type was set to Frontal collision, and the crush extent was set to Zone 1. The explanatory power of the predictive model was confirmed using the concordance statistics (c-statistics) and the area under the curve (AUC) value. Multicollinearity was verified, and a Hosmer–Lemeshow goodness-of-fit test was performed to determine fitness. The most common method for finding the adjusted odds ratio (hereinafter, OR) was logistic regression. In this method, the logistic coefficients were the logarithm of the respective OR. This regression can be run with several covariates in the logistic model. Each coefficient provides ln (OR) for that factor and this was automatically adjusted for the other covariates in the model [27].

#### 2.3.2. External Validation Analysis

After developing the predictive model, in order to develop a generalized predictive model, the external validation was evaluated by applying the predictive model to the 2019 data of the same data structure as the data at the time of model development in the KIDAS database. The diagnosed TBI and the predicted TBI categorized in the receiver operating characteristic (ROC) analysis were specified. When the predictive model was performed external validation, four outcomes were derived from it. A true positive (TP) was an outcome where the model correctly predicted the positive class. Similarly, a true negative (TN) was an outcome where the model correctly predicts the negative class. In this study, TP and TN measurements were associated with cases of accurate injury prediction using current models. On the contrary, a false positive (FP) was an outcome where the model incorrectly predicts the positive class, and a false negative (FN) was an outcome where the model incorrectly predicts the negative class. FN measurement was associated with underprediction, which was when non-TBI was expected but there was actually TBI. FP measurement was associated with overprediction, which was when there was a prediction of TBI that was actually non-TBI. Sensitivity was defined as the number of TP divided by the number of TP plus FN and specificity was defined as the number of TN divided by the number of TN plus FP.

#### 2.3.3. Statistical Analysis

Statistical analysis was performed using SPSS (Version 25.0, IBM Inc., Chicago, IL, USA). Comparisons among nominal variables were expressed as frequency and percentage using the chi-squared test or Fisher’s exact test. Comparisons among continuous variables were recorded as means and standard deviations using the *t*-test or analysis of variance (ANOVA) according to the number of independent variables. Variables that did not follow the normal distribution such as the AIS and ISS were recorded in the median and quartile ranges using the Mann–Whitney U test or the Kruskal–Wallis H test. Multiple logistic regression was used to identify factors affecting the traumatic brain injuries in the elderly MVOs, and *p* values less than 0.05 were judged to be statistically significant. The explanatory power, cut-off value calculation, and a receiver operating characteristic (ROC) curve of the constructed predictive model were visualized in MedCalc version 11.2.1.0 (MedCalc Software, Mariakerke, Belgium).

## 3. Results

### 3.1. General Characteristics

Table 1 shows the general characteristics of TBI in the elderly MVOs. There were statistically significant differences in the collision type. In the TBI group, there were 165 frontal collision (46.2%), 36 lateral-nearside collision (10.1%), 23 lateral-farside collision (6.4%), 37 rear-end collision (10.4%), 54 rollover (15.1%), and 42 multiple collisions (11.8%). In the non-TBI group, there were 259 frontal collision (55.7%), 35 lateral-nearside collision (7.5%), 37 lateral-farside collision (8.0%), 39 rear-end collision (8.4%), 60 rollover (12.9%), and 35 multiple collisions (7.5%) (*p* = 0.049). In addition, there was a statistically significant in mental status. In the TBI group, there were 289 in alert (85.8%), 24 in verbal response (7.1%), 12 in pain response (3.6%), and 12 in unresponsive (3.6%). In the non-TBI group, there were 368 in alert (92.2%), 16 in verbal response (4.0%), 1 in pain response (0.3%), and 14 in unresponsive (3.5%) (*p* = 0.001). 

The rate of the fastened seatbelt in the TBI group (63.6%) was lower than the non-TBI group (72.7%) (*p* = 0.008). In addition, there was a statistically significant difference between the TBI group and the non-TBI group in the median MAIS [lower quartile and upper quartile] to 2 [2–3] vs. 2 [1–3], (*p* < 0.001). In addition, there was a statistically significant difference between the TBI group and the non-TBI group in the median ISS [lower quartile and upper quartile] to 6 [3–13] vs. 5 [2–12], (*p* < 0.001).

### 3.2. Factors Affecting TBI in the Elderly MVOs

Table 2 shows the factors affecting TBI in the elderly. By the multiple logistic regression analysis (χ_2_ = 7.123, *p* value: 0.523, Hosmer–Lemeshow test), the derived model was good-to-fit. Compared to frontal collisions, the OR [95% CI] for TBI in the elderly was 1.833 [1.077–3.119] times greater in rear-end collisions and 1.897 [1.136–3.167] in multiple collisions; neither OR was statistically significant. In addition, the OR of TBI in the elderly in the unfastened seatbelt was 1.677 [1.215–2.315] times higher related to the fastened.

### 3.3. Logistic Regression Model

Table 3 shows the beta coefficients for the variables retained in the logistic regression model. The indicators were adopted sex, vehicle type, seating position, seatbelt status, collision type, and crush extent. The standard indices, namely Male, Sedan, Driver seat, Fastened, Frontal collision, and Crush extent 1, were applied as 0 when the condition was applied. The derived model was statistically meaningful according to Hosmer–Lemeshow goodness-of-fit test results (*p* = 0.523).

Using the results in Table 3, the predictive model to determine the presence of TBI in elderly MVOs is as shown in Equation (2).
(2)$$P\left(TBI\;incidence\;in\;the\;elderly\;MVOs\right)=\frac{e^{A}}{1+e^{A}}$$
where *A* = −0.561 − 0.076 × Sex_[Female]_ − 0.245 × Vehicle_[SUV]_ − 0.198 × Vehicle_[Light_truck]_ − 0.293 × Vehicle_[Van]_ + 0.125 × Seating_position_[passenger]_ − 0.123 × Seating_position_[2nd-row left passenger]_ − 0.765 × Seating_position_[2nd-row right passenger]_ + 0.517 × Seatbelt _[Unfastened]_ + 0.468 × Collision_type_[Lateral-nearside]_ + 0.118 × Collision_type_[Lateral-farrside]_ + 0.606 × Collision_type_[Rear-end]_ + 0.393 × Collision_type_[Rollover]_ + 0.640 × Collision_type_[Multiple]_ + 0.031 × Crush_extent.

The accuracy of the predictive model represented by the ROC curve is related to the area under the ROC curve (AUC). Therefore, the larger the AUC, the more accurate the model is considered. The predictive model was 60.8% (c-statistics: 0.608). The cut-off value of the derived predictive model from Equation (2) was 0.4832 (sensitivity 41.7%; specificity 76.8%). When the value calculated using the above equation exceeded 0.4832, this determined a group with TBI (Table 4).

### 3.4. External Validation of the Model

Data from the KIDAS database, collected in 2019, were used to analyze the external validity of the derived prediction model. Among the total 156 MVOs, external validity analysis was conducted with 43 MVOs corresponding to the indicators derived from the logistic regression model. Table 5 shows that the sensitivity, specificity, and accuracy of the TBI MVOs actually diagnosed and the TBI MVOs predicted by the predictive model were analyzed to be 0.500, 0.730, and 0.698, respectively.

## 4. Discussion

### 4.1. Methodology

In this study, the collision type was classified into frontal, lateral-nearside, lateral-farside, rear-end, rollover, and multiple collisions (having two or more collision directions) using general area of deformation in the CDC code. In addition, vehicle damage information due to the amount of change in the velocity was expressed as the crush extent, the 7th column of the CDC code [21].

In the derived prediction model for TBI in elderly MVOs from this study, the crush extent, the seventh column of the CDC code, was selected as one of the factors. Clearly, the rate of change in the velocity was the most accurate indicator of the severity of the MVCs. However, it was difficult to know the rate of change in the velocity in the actual emergency scene. Alternately, vehicle deformation was usually used as a substitute indicator for evaluating the severity of an MVCs. Guidelines for field triage of injured patients should be transferred to a severe trauma center when the in-vehicle interior or roof breakage was greater than 30 cm (12 inches) or intrusion into the occupant area was 45 cm (18 inches) in the MVCs [17]. However, it was difficult for the emergency medical technicians to measure the deformation amount of the vehicle numerically in the actual field. In addition, in Korea, due to legal issues, there was another limited investigation environment. The investigation system of MVCs in Korea is not unified, and the police, medical team, and investigation team are separated. In addition, since the police report cannot be obtained by the personal information protection law, there is a limit in which the collision speed value is unknown.

In conclusion, in this study, the crush extent of the CDC code, which can be easily distinguished from vehicle damage in the actual field, was used as an alternative index to evaluate the severity of crashes. Moreover, to compensate for this weakness, considering that the relationship between the maximum crush extent and the collision speed is linear, this value was selected as an influencing factor of the prediction model for TBI in elderly MVOs [28,29].

### 4.2. General Characteristics

According to the collision type, the TBI incidence in most collision types, except for the frontal collision and lateral-farside collision, was higher than in the non-TBI. Neyens and Boyle presented that the rate of angular crash in the TBI group (20.8%) was lower than in the controls (25.4%) and the rate of rear-crash in the TBI group (19.1%) was lower than in the controls (36.8%). Angular crashes were defined as involving two or more vehicles that are not traveling in parallel directions to each other prior to the crash. It seemed to be similar to the concepts of lateral-nearside collision and lateral-farside collision in this study. However, the results of this study were different from Neyens’s study [16].

In case of the TBI in MVOs, the rate of the fastened seatbelt was lower than in the non-TBI in MVOs. In addition, the ISS of the TBI group was higher than of the non-TBI group. This is related to what Mayrose reported, that persons who fail to use their seatbelt experience a far greater force of impact in MVCs than persons who fasten their seatbelts. The greatest risk that could arise if the seatbelt was not worn when the vehicle crashes is the possibility of a secondary impact. It was reported from collision experiments that motorists who unfastened their seatbelt would be propelled forward and upward due to inertia, causing their chest to inevitably strike the steering wheel, and their forehead to strike the glass of the windshield, resulting in injury [30].

### 4.3. Logistic Multiple Regression Analysis

Table 2 presented the logistical regression analysis used to determine the cause of TBI in MVCs by selecting sex, vehicle type, seating position, seatbelt status, collision type, and crush extent.

The OR (95% confidence interval, CI) for the incidence of TBI with an unfastened seatbelt was 1.677 (1.215–2.315) times relative to a fastened seatbelt. What the seatbelt had a preventive effect in this study was similar to the previous study by Evans et al. It was presented that primary restraint devices, such as the 3-point seatbelt system, are approximately 42% to 45% effective at preventing fatal injuries and approximately 65% effective in preventing serious injuries among MVOs [31].

As a result of the external validity analysis, the specificity was higher than the sensitivity of the predictive model. The predictive model of this study was intended to determine the occurrence of TBI in the elderly MVOs and, though the probability of occurrence was high whereas the risk was low, we would consider this model may be improved through further research.

## 5. Conclusions

In this study, we suggested a model that could predict the traumatic brain injury in the elderly of actual vehicle MVOs based on the real-world evidence in Korea. It is expected that there will be more factors affecting TBI in the elderly MVOs, and further studies are needed to improve the predictive model to reflect additional indicators. For example, there has been focus on head injury criteria thresholds such as peak linear acceleration (PLA), head injury criterion (HIC), and the evolution of the Wayne state tolerance curve (WSTC) in mild brain traumatic injury [6]. In the KIDAS database used in this study, there were no continuous values related to vehicle dynamics such as speed, acceleration, change in speed (delta-v), and impulse, and these were not applied to the prediction model. Future studies are required to improve the predictive model by reflecting additional indicators.

## 6. Limitations

KIDAS is a database collected through real-world investigations of the MVOs who visited the emergency departments of five regional emergency centers. The critical limitation of this study was that the number of samples was too small to create a predictive model. In addition, there were many incomplete indicators in Korea due to the Personal Information Protection Act, because there was still an insufficient system to secure field reports, field photos, and vehicle photos during the investigation of crashes. This resulted in limitation of our study’s statistical power as conditions for exclusion occurred. However, this study was meaningful in that it suggested a model that could predict traumatic brain injury in the elderly MVOs of actual vehicle MVCs based on real-world evidence in Korea.

## Figures and Tables

**Figure 1 ijerph-18-03975-f001:**
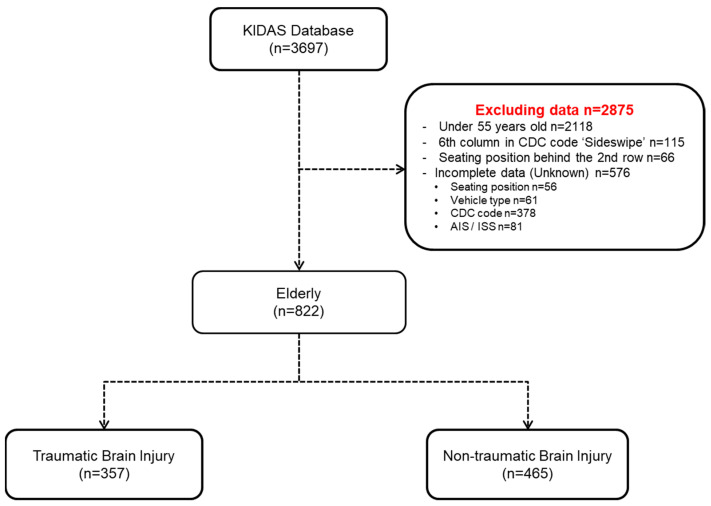
Flowchart of classifying the data. KIDAS: Korean In-Depth Accident Study, CDC: Collision Deformation Classification, AIS: Abbreviated Injury Scale, ISS: Injury Severity Score.

**Table 1 ijerph-18-03975-t001:** General characteristics of elderly motor vehicle occupants (MVOs).

Variables	Total (*n* = 822)	TBI (*n* = 357)	Non-TBI (*n* = 465)	*p* Value
Sex, *n* (%)				0.646 *
Male	505 (61.4)	223 (62.5)	282 (60.6)	
Female	317 (38.6)	134 (37.5)	183 (39.4)	
Age (years), mean ± SD	63.53 ± 7.25	63.44 ± 7.46	63.60 ± 7.10	0.764
Height (cm), mean ± SD	*n* = 575	*n* = 237	*n* = 338	0.113
	163.19 ± 9.96	163.97 ± 7.97	162.64 ± 11.12	
Weight (kg), mean ± SD	*n* = 576	*n* = 238	*n* = 338	0.838
	64.04 ± 10.13	64.14 ± 9.98	63.97 ± 10.25	
BMI (kg/m^2^), mean ± SD	*n* = 572	*n* = 236	*n* = 336	0.461
	23.93 ± 3.02	23.82 ± 2.90	24.01 ± 3.10	
Vehicle type, *n* (%)				0.741
Sedan	399 (48.5)	179 (50.1)	220 (47.3)	
SUV	161 (19.6)	66 (19.6)	95 (20.4)	
Light truck	174 (21.2)	77 (21.2)	97 (20.9)	
Van	88 (10.7)	35 (10.7)	53 (11.4)	
Collision type, *n* (%)				0.049
Frontal collision	424 (51.6)	165 (46.2)	259 (55.7)	
Lateral-nearside collision	71 (8.6)	36 (10.1)	35 (7.5)	
Lateral-farside collision	60 (7.3)	23 (6.4)	37 (8.0)	
Rear-end collision	76 (9.2)	37 (10.4)	39 (8.4)	
Rollover	114 (13.9)	54 (15.1)	60 (12.9)	
Multiple collisions	77 (9.4)	42 (11.8)	35 (7.5)	
Fastened seatbelt, *n* (%)	*n* = 796	*n* = 349	*n* = 447	0.008
	547 (66.5)	222 (63.6)	325 (72.7)	
Deployed frontal airbag, *n* (%)	*n* = 610	*n* = 279	*n* = 331	0.356
	154 (25.2)	65 (23.3)	89 (26.9)	
Seating position, *n* (%)				0.201
Driver	521 (63.4)	225 (63.0)	296 (63.7)	
Passenger	202 (24.6)	94 (26.3)	108 (23.2)	
2nd-row left	39 (4.7)	19 (5.3)	20 (4.3)	
2nd-row right	60 (7.3)	19 (5.3)	41 (8.8)	
Seating row, *n* (%)				0.331 *
1st-row	723 (88.0)	319 (89.4)	404 (86.9)	
2nd-row	99 (12.0)	38 (10.6)	61 (13.1)	
Crush extent (CE), mean ± SD	3.38 ± 1.79	3.43 ± 1.81	3.34 ± 1.79	0.586
Crush extent (CE) zone, *n* (%)				0.570
Zone 1 (Extent 1–3)	537 (65.3)	233 (65.3)	304 (65.4)	
Zone 2 (Extent 4–6)	220 (26.8)	92 (25.8)	128 (27.5)	
Zone 3 (Extent 7–9)	65 (7.9)	32 (9.0)	33 (7.1)	
Alcohol, *n* (%)	*n* = 584	*n* = 259	*n* = 325	0.210 *
No	554 (94.9)	248 (95.8)	306 (94.2)	
Yes	30 (5.1)	11 (4.2)	19 (5.8)	
Mental status, *n* (%)	*n* = 736	*n* = 289	*n* = 368	0.001
Alert	657 (89.3)	289 (85.8)	368 (92.2)	
Verbal response	40 (5.4)	24 (7.1)	16 (4.0)	
Pain response	13 (1.8)	12 (3.6)	1 (0.3)	
Unresponsive	26 (3.5)	12 (3.6)	14 (3.5)	
Result of emergency room, *n* (%)	*n* = 750	*n* = 336	*n* = 414	0.143
Discharge	120 (16.0)	61 (18.2)	59 (14.3)	
Transfer	146 (19.5)	63 (18.8)	83 (20.0)	
Ward admission	357 (47.6)	146 (43.5)	211 (51.0)	
ICU admission	92 (12.3)	47 (14.0)	45 (10.9)	
Expired	35 (4.7)	19 (5.7)	16 (3.9)	
Result of admission, *n* (%)	*n* = 314	*n* = 151	*n* = 163	0.115
Discharge	259 (82.5)	119 (78.8)	140 (85.9)	
Transfer	47 (15.0)	29 (19.2)	18 (11.0)	
Expired	8 (2.5)	3 (2.0)	5 (3.1)	
MAIS, median [IQR]	2 [1–3]	2 [2–3]	2 [1–3]	<0.001
ISS, median [IQR]	5 [2–13]	6 [3–13]	5 [2–12]	<0.001

SD: standard deviation; IQR: interquartile range; BMI: body mass index; MAIS: maximum abbreviated injury scale; ISS: injury severity score; * *p* value by Fisher’s exact test.

**Table 2 ijerph-18-03975-t002:** Factors affecting the presence of traumatic brain injury (TBI) in the elderly.

Variables	Univariate	Multivariate
Sex, *n* (%)		
Male	Reference	Reference
Female	0.926 (0.697–1.230)	0.927 (0.663–1.296)
Age (year)	0.997 (0.978–1.016)	
Height (cm)	1.016 (0.996–1.035)	
Weight (kg),	1.002 (0.985–1.018)	
BMI (kg/m^2^)	0.979 (0.927–1.035)	
Vehicle type		
Sedan	Reference	Reference
SUV	0.854 (0.589–1.237)	0.783 (0.528–1.161)
Light truck	0.976 (0.682–1.396)	0.821 (0.547–1.231)
Van	0.812 (0.507–1.299)	0.746 (0.452–1.232
Collision type		
Frontal collision	Reference	Reference
Lateral-nearside collision	1.615 (0.975–2.674)	1.597 (0.938–2.718)
Lateral-farside collision	0.976 (0.560–1.701)	1.125 (0.629–2.014)
Rear-end collision	1.489 (0.912–2.432)	1.833 (1.077–3.119)
Rollover	1.413 (0.932–2.142)	1.481 (0.959–2.288)
Multiple collision	1.884 (1.155–3.072)	1.897 (1.136–3.167)
Seatbelt		
Unfasten (vs. Fasten—Ref)	1.524 (1.127–2.060)	1.677 (1.215–2.315)
Frontal airbag		
Non-deployment (vs. Deployment—Ref)	1.211 (0.837–1.751)	
Curtain airbag, *n* (%)		
Non-deployment (vs. Deployment—Ref)	1.158 (0.449–2.982)	
Seating position, *n* (%)		
Driver	Reference	Reference
Passenger	1.145 (0.826–1.587)	1.134 (0.783–1.640)
2nd-row left	1.250 (0.652–2.397)	0.884 (0.419–1.868)
2nd-row right	0.610 (0.344–1.079)	0.465 (0.941–1.129)
Seating row, *n* (%)		
1st-row	Reference	
2nd-row	0.789 (0.513–1.214)	
Crush extent (CE)	1.026 (0.950–1.107)	1.031 (0.941–1.129)
Crush extent (CE) zone, *n* (%)		
Zone 1	Reference	
Zone 2	0.938 (0.683–1.288)	
Zone 3	1.265 (0.756–2.118)	
Hosmer–Lemeshow: λ^2^ = 7.123, *p* = 0.523, Nagelkerke R^2^ = 0.050		

**Table 3 ijerph-18-03975-t003:** Beta coefficients for the variables retained in the logistic regression model.

Variables		β	SE	Wald	*p* Value
Intercept		−0.561	0.222	6.389	0.011
Sex	Female (vs. Male)	−0.076	0.171	0.197	0.657
Vehicle type	Sedan	Reference		2.529	0.470
	SUV	−0.245	0.201	1.487	0.223
	Light truck	−0.198	0.207	0.912	0.339
	Van	−0.293	0.256	1.314	0.252
Seating position	Driver	Reference			0.070
	Front Right Passenger	0.125	0.189	0.442	0.506
	Second Left Passenger	−0.123	0.382	0.104	0.747
	Second Right Passenger	−0.765	0.333	5.271	0.022
Seatbelt status	Unfastened (vs Fasten)	0.517	0.164	9.902	0.002
Collision type	Frontal collision	Reference		7.060	0.037
	Lateral-Nearside collision	0.468	0.271	2.974	0.085
	Lateral-farside collision	0.118	0.297	0.158	0.691
	Rear-end collision	0.606	0.271	4.985	0.026
	Rollover	0.393	0.222	3.134	0.077
	Multiple collisions	0.640	0.262	5.991	0.014
Crush extent (increased 1 unit)		0.031	0.046	0.435	0.510

**Table 4 ijerph-18-03975-t004:** Explanatory power of the model to determine TBI in elderly MVOs.

c-Statistics (95% CI)	Cut-Off Value	Sensitivity	Specificity
60.8% (57.4%, 64.2%)	0.4832	0.417	0.768

**Table 5 ijerph-18-03975-t005:** Verification results of the diagnosed TBI and predicted TBI.

TBI in the Elderly MVOs		Diagnosed Condition	
		TBI	non-TBI
**Predicted condition**	**TBI**	3 (TP: True Positive)	10 (FP: False Positive)
	**non-TBI**	3 (FN: False Negative)	27 (TN: True Negative)
Sensitivity: 0.500 (TP/(TP + FN)), Specificity: 0.730 (TN/(FP + TN)), Accuracy: 0.698 ((TP + TN)/All)			

## Data Availability

The datasets analyzed during the current study are not yet publicly available but are available from the corresponding author on reasonable request.

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
