# Peer review of "A Predictive Model to Analyze the Factors Affecting the Presence of Traumatic Brain Injury in the Elderly Occupants of Motor Vehicle Crashes Based on Korean In-Depth Accident Study (KIDAS) Database"

_ijerph, 2021, doi:10.3390/ijerph18083975_

Round 1
Reviewer 1 Report
I kindly suggest 2 publications from the EU REHABILAID project:
1) Trauma https://doi.org/10.1177/1460408616677564
2) Injury V 48, Issue 2, February 2017, pp 297-306
Author Response
리뷰어에 대한 답변 댓글 1 개
포인트 1 :
EU REHABILAID 프로젝트에서 2 개의 출판물을 제안합니다.
1) 트라우마 https://doi.org/10.1177/1460408616677564
2) Injury V 48, Issue 2, 2017 년 2 월, pp 297-306
응답 1 :
Thank you for recommending brilliant publications for me.
I read the articles you recommended and wrote the contents of the introduction to my manuscript.
Please review the points indicated in red letters.
[Page 2. Line 68-69] However, in recent studies, it has been emphasized that the health policy for injury prevention should be focused on groups with high risk of injury [12].
[Page 2. Line 77-80] In a cohort study conducted in Europe, it was reported that the possibility of returning to society can be increased if severe damage is quickly predicted and rehabilitation and psychological treatment are carried out quickly for victims of motor vehicle crashes [16].
[페이지 13. Line 415-418] Papadakaki, M .; 매사추세츠 주 Stamouli; Ferraro, OE; Orsi, C .; Otte, D .; Tzamalouka, G .; Von-der-Geest, M .; Lajunen, T .; Ozkan, T .; Morandi, A .; Kotsyfos, V .; Chliaoutakis, J. 입원 비용 및 도로 교통 사고로 인한 부상으로 인한 직간접 적 경제적 손실 추정치 : 유럽 3 개국에서 실시한 1 년 코호트 연구 결과 (REHABILAID 프로젝트). 트라우마, 2017, 19 (4), 264-276.
[13 페이지 425-427 행] Papadakaki, M .; Ferraro, O E .; Orsi, C .; Otte, D .; Tzamalouka, G .; Von-der-Geest, M., Lajunen, T .; Ozkan, T .; Morandi, A .; Sarris, M .; Pierrakos, G .; Chliaoutakis, J. 도로 교통 사고로 심각한 부상을 입은 환자의 심리적 고통 및 신체 장애 : 유럽 3 개국에서 실시한 1 년 코호트 연구 결과. 부상, 2017, 48 (2), 297-306.

Reviewer 2 Report
The study is interesting, and although similar models already exist in the literature, the authors clearly identify the novelty points introduced by their work. However, the authors need to address the following issues:
- Equation 1 needs revision. Clearly something is wrong, but some editing/formatting is enough to correct it
- Please comment on the higher p-values and lower R2. These can undermine the model validity
- Equation 2 + line 267. Again, some issues with equations
- Although a proper discussion is presented, the conclusion section could be improved. Consider the contributions to the literature
- The introduction and literature review need improvement regarding TBI. A suggestion: 1177/0954411915592906
Author Response
Response to Reviewer 2 Comments
Thank you for your sincere review of my paper.
I have tried to answer your review points in a polite and detailed.
Please check the points highlighted in green in the manuscript.
Point 1: Equation 1 needs revision. Clearly something is wrong, but some editing/formatting is enough to correct it.
Response 1: As a result of checking Equation 1, there was no problem with the formula. However, it seems that I read it confusingly because I did not express the formula in general. So I modified it to the format of the formula commonly expressed in the manuscript.
[Page 4~5. Line 176~181]
|
|
(1) |
Where P(Y=1) is the probability of an occurrence of the TBI in the elderly MVOs and A is the partial regression coefficient of each x, estimated using the maximum-likelihood method.
Point 2: Please comment on the higher p-values and lower R2. These can undermine the model validity.
Response 2: R2 (R-square) refers to the explanatory power of the model. In the linear model, it is possible to use R2 to express the explanatory power of the model, and to calculate the psedo- R2 value in the logistic regression model. So, I thought that there was no problem with the fit of the model with p-value = 0.523 as a result of ‘Hosmer-Lemeshow x2-test’ specified at the bottom of Table 2.
Point 3: Equation 2 + line 267. Again, some issues with equations.
Response 3: In the same pattern as Equation 1, there was no problem with the formula. However, it seems that I read it confusingly because I did not express the formula in general. So I modified it to the format of the formula commonly expressed in the manuscript.
[Page 9. Line 272]
|
|
(2) |
Point 4: Although a proper discussion is presented, the conclusion section could be improved. Consider the contributions to the literature.
Response 4: The conclusions have been revised concisely, and a new limitation section has been added to describe the weak points in this study, points to be supplemented in future studies, and the implications of this study.
[Page 11. Line 355-377]
- Conclusions
In this study, we suggested a model that could predict the traumatic brain injury in the elderly of actual vehicle MVOs based on the real-world evidence in Korea. It is expected that there will be more factors affecting the TBI in the elderly MVOs, and further studies need to improve the predictive model to reflect additional indicators. For example, it has been focused on the head injury criteria thresholds like the peak linear acceleration (PLA), the head injury criterion (HIC), and the evolution of the Wayne state tolerance curve (WSTC) in the mild brain traumatic injury. In the KIDAS DB used in this study, there were no continuous values related to vehicle dynamics such as speed, acceleration, change in speed (delta-v), and impulse, and these were not applied to the prediction model. Future studies are required to improve the predictive model by reflecting additional indicators.
- Limitations
KIDAS is a database collected through real-world investigations of the MVOs who visit the emergency departments of five regional emergency centers. The critical limitation of this study was that the number of samples was too small to create a predictive model. In addition, there were many incomplete indicators in Korea due to the Personal Information Protection Act, because there was still an insufficient system to secure field reports, field photos, and vehicle photos during the investigation of crashes. It made this study be limit our statistical power as conditions for exclusion occurred. However, this study was meaningful in that it suggested a model that could predict the traumatic brain injury in the elderly of actual vehicle MVOs based on the real-world evidence in Korea.
Point 5: The introduction and literature review need improvement regarding TBI. A suggestion: 1177/0954411915592906
Response 5: Thank you for recommending excellent publication for me. I read the article you recommended and wrote the contents of the introduction and discussion to my manuscript.
[Page 2. Line 57~59]
Traumatic brain injury (TBI), also known as intracranial injury, occurs when an external force injures the brain. Fernandes and Sousa introduced head injuries and their mechanisms, and reviewed in detail the thresholds for head injuries and the criteria for each head injury [6]. The TBI can be classified based on severity, mechanism, or other features.
[Page 12. Line 359~366]
For example, it has been focused on the head injury criteria thresholds like the peak linear acceleration (PLA), the head injury criterion (HIC), and the evolution of the Wayne state tolerance curve (WSTC) in the mild brain traumatic injury. In the KIDAS DB used in this study, there were no continuous values related to vehicle dynamics such as speed, acceleration, change in speed (delta-v), and impulse, and these were not applied to the prediction model. Future studies are required to improve the predictive model by reflecting additional indicators.
[Page 12. Line 402~403]
- Fernandes, F.A.; Sousa, R.J.A.D. Head injury predictors in sports trauma–a state-of-the-art review. Proceedings of the Institution of Mechanical Engineers, Part H: Eng. Med., 2015, 229(8), 592-608.

Reviewer 3 Report
The paper presents a statistical study of factors associated with TBI amongst >55 years occupants in Korean motor vehicle crashes. A logistic regression analysis is used to find significant factors and build a predictive model. The results indicate that few variables are statistically significant, thus the power and significance of the model and the outputs are relatively low, and there is little to substantively contribute to the topic. Nonetheless, the statistical methods and their interpretations are generally sound, although I had a few clarifications/queries as below:
1. lines 244 – 250 – this is Methods not Results
2. Table 2 – please clarify that the “adjusted” column relates to the logistic regression model eg in the table caption
3. for clarity it should be stated that the values in Table 3 relate to the “adjusted” model shown in Table 2
4. it would be useful to add the p values for the “adjusted” model in Table 2
5. line 253 – states that neither OR was statistically significant – according to what measure? Usually for 95% CI when both values are either above or below 1.0 the OR is statistically significant
6. line 257 and logistic regression model – please explain the retention criteria for variables in the logistic regression model, typically (but not necessarily) only variables with p<0.05 are retained but this is not the case here
7. please briefly state the definitions for beta, SE and Wald used in Table 3
8. line 271 – please explain how the cutoff value of 0.4832 was derived – usually this is selected by the modeller to achieve desired values of sensitivity and specificity relevant to the application
9. Section 4.3 - it is highly unusual that age and crush extent were not statistically significant, this could be elaborated in this discussion
10. Section 4.3 – in Table 2 it seems rear-end and multiple collisions were statistically significant (or nearly statistically significant) - this could be elaborated in this discussion
11. Conclusions – when the title and aim of the study is to build a predictive model, it seems to me to be inappropriate to conclude that “the number of samples was too small to create a predictive model”. You have successfully created a model, its just that it has low power.
12. Currently the Conclusions section is actually a Limitations section. I suggest you call it Limitations, then add a proper Conclusion
Author Response
Response to Reviewer 3 Comments
Thank you for your sincere review of my paper.
I have tried to answer your review points in a polite and detailed.
Please check the points highlighted in yellow in the manuscript.
Point 1: lines 244 – 250 – this is Methods not Results.
Response 1: Thank you for your comment. As your comment, I rewrote the sentences to Methods
[Page 5. Line 189-193]
The most common method for finding the adjusted odds ratio (hereinafter, OR) was logistic regression. In this method, the logistic coefficients were the logarithm of the respective ORs. This regression can be run with several covariates in the logistic model. Each coefficient provides ln (OR) for that factor and this was automatically adjusted for the other covariates in the model [25].
Point 2: Table 2 – please clarify that the “adjusted” column relates to the logistic regression model eg in the table caption.
Response 2: Crude OR was a calculation of the effect of each univariate on TBI, and Adjusted OR was a multivariate that calculated the effect of correcting the remaining variables for one variable. I modified the crude OR and adjusted OR with the commonly used univariate and multivariate respectively.
[Page 8. Line 253]
|
Crude OR -> Uni-Variate |
Adjusted OR -> Multi-variate |
Point 3: for clarity it should be stated that the values in Table 3 relate to the “adjusted” model shown in Table 2.
Response 3: Thank you for your good comment. As the table below, there were beta coefficient, SE, Wald, Degree of freedom, p-value, Exp(B) in the table. In the context of this paper, I put the analysis values of univariate and multivariate in Table 2, and in Table 3, I thought and described it as deriving a logistic regression model using the analysis values of multivariate. Probably reflecting your comment, combining Table 2 and Table 3 into one would be a good revision direction.
Point 4: it would be useful to add the p values for the “adjusted” model in Table 2.
Response 4: As your comment, I added the p-values for the “adjusted” model in Table 2.
[Page 9. Line 269]
|
Variables |
β |
SE |
Wald |
p-value |
|
|
Intercept |
-0.561 |
0.222 |
6.389 |
0.011 |
|
|
Sex |
Female (vs Male) |
-0.076 |
0.171 |
0.197 |
0.657 |
|
Vehicle type |
Sedan |
Reference |
|
2.529 |
0.470 |
|
SUV |
-0.245 |
0.201 |
1.487 |
0.223 |
|
|
Light truck |
-0.198 |
0.207 |
0.912 |
0.339 |
|
|
Van |
-0.293 |
0.256 |
1.314 |
0.252 |
|
|
Seating position |
Driver |
Reference |
|
|
0.070 |
|
Front Right Passenger |
0.125 |
0.189 |
0.442 |
0.506 |
|
|
Second Left Passenger |
-0.123 |
0.382 |
0.104 |
0.747 |
|
|
Second Right Passenger |
-0.765 |
0.333 |
5.271 |
0.022 |
|
|
Seatbelt status |
Unfastened (vs Fasten) |
0.517 |
0.164 |
9.902 |
0.002 |
|
Collision type |
Frontal collision |
Reference |
|
7.060 |
0.037 |
|
Lateral-Nearside collision |
0.468 |
0.271 |
2.974 |
0.085 |
|
|
Lateral-farside collision |
0.118 |
0.297 |
0.158 |
0.691 |
|
|
Rear-end collision |
0.606 |
0.271 |
4.985 |
0.026 |
|
|
Rollover |
0.393 |
0.222 |
3.134 |
0.077 |
|
|
Multiple collisions |
0.640 |
0.262 |
5.991 |
0.014 |
|
|
Crush extent (increased 1 unit) |
0.031 |
0.046 |
0.435 |
0.510 |
|
Point 5: line 253 – states that neither OR was statistically significant – according to what measure? Usually for 95% CI when both values are either above or below 1.0 the OR is statistically significant.
Response 5: It was used to create a predictive model because it was determined that it was clinically meaningful even though the statistical significance was slightly lower as a single variable.
Point 6: line 257 and logistic regression model – please explain the retention criteria for variables in the logistic regression model, typically (but not necessarily) only variables with p<0.05 are retained but this is not the case here.
Response 6:
☞ In this study, I adopted variables using logistic regression variable selection methods. Especially, I treated the variables as forced-in. Additionally, I tried to explain how I chose the variables by using logistic regression variable selection methods as below.
☞ When proceeding with the ‘Forward Selection (Wald)’ method, three variables were adopted in three stages. (Variables: Collision type -> Collision type, Seatbelt status -> Collision type, Seatbelt status, Vehicle type)
☞ Also, when proceeding with the Backward Elimination (Wald) method, three variables were adopted in 10 steps. (Variables: Sex, Age, Height, Weight, BMI, Vehicle type, Crush extent, Collision type, Seating row, Seatbelt status, Frontal airbag, Curtain airbag -> Curtain airbag (eliminated) -> Curtain airbag, Age (eliminated) -> Curtain airbag, Age, Frontal airbag (eliminated) -> ``` -> Collision type, Seatbelt status, Vehicle type (adopted)
☞ By using logistic regression variable selection methods, forward selection or backward elimination, three indicators were adopted first. [Collision type, Seatbelt status, Vehicle type]
☞ Moreover, I thought sex and age are basically human factors that affect TBI. However, since this paper was written for the elderly MVOs, I adopted sex as a predictive model factor. [Sex]
☞ Finally, I thought that there was a correlation between the collision speed and TBI. Therefore, the crush extent, which could estimate the collision speed, was adopted as a predictive model factor. [Crush extent]
Point 7: please briefly state the definitions for beta, SE and Wald used in Table 3
Response 7: The beta is the weight factor before each explanatory variable in the predictive model. SE means the standard error of the estimated beta value. Wald is a test-statistic, a statistic to test whether the estimated beta value is statistically zero or not, and calculates the p-value using the degree of freedom value.
Point 8: line 271 – please explain how the cutoff value of 0.4832 was derived – usually this is selected by the modeller to achieve desired values of sensitivity and specificity relevant to the application
Response 8: In general, the predictable value is based on 0.5. The contact point when the area of the curve of the prediction model equation is the largest AUC is 0.4832.
Point 9: Section 4.3 - it is highly unusual that age and crush extent were not statistically significant, this could be elaborated in this discussion
Response 9: In your opinion, age is an influencing factor when it comes to group comparisons between people in ordinary papers. However, in this paper, I think that age was not significant as a predictive model factor because the target group had adopted the MVO for the elderly over 55 years of age from the beginning. In addition, the crush extent is also from 1 to 9 according to the CDC code. To see a more significant difference, it is expected that if the controlled definition was defined as a crush extent zone by making some groups, it would have come out significantly.
Point 11: Conclusions – when the title and aim of the study is to build a predictive model, it seems to me to be inappropriate to conclude that “the number of samples was too small to create a predictive model”. You have successfully created a model, its just that it has low power.
Response 11: As your comment, I rewrote the conclusions have been revised with another expression, concisely.
Point 12: Currently the Conclusions section is actually a Limitations section. I suggest you call it Limitations, then add a proper Conclusion
Response 12: The conclusions have been revised concisely, and a new limitation section has been added to describe the weak points in this study, points to be supplemented in future studies, and the implications of this study.
[Page 11. Line 355-377]
- Conclusions
In this study, we suggested a model that could predict the traumatic brain injury in the elderly of actual vehicle MVOs based on the real-world evidence in Korea. It is expected that there will be more factors affecting the TBI in the elderly MVOs, and further studies need to improve the predictive model to reflect additional indicators. For example, it has been focused on the head injury criteria thresholds like the peak linear acceleration (PLA), the head injury criterion (HIC), and the evolution of the Wayne state tolerance curve (WSTC) in the mild brain traumatic injury. In the KIDAS DB used in this study, there were no continuous values related to vehicle dynamics such as speed, acceleration, change in speed (delta-v), and impulse, and these were not applied to the prediction model. Future studies are required to improve the predictive model by reflecting additional indicators.
- Limitations
KIDAS is a database collected through real-world investigations of the MVOs who visit the emergency departments of five regional emergency centers. The critical limitation of this study was that the number of samples was too small to create a predictive model. In addition, there were many incomplete indicators in Korea due to the Personal Information Protection Act, because there was still an insufficient system to secure field reports, field photos, and vehicle photos during the investigation of crashes. It made this study be limit our statistical power as conditions for exclusion occurred. However, this study was meaningful in that it suggested a model that could predict the traumatic brain injury in the elderly of actual vehicle MVOs based on the real-world evidence in Korea.

Round 2
Reviewer 2 Report
All my comments were properly addressed. No further remarks.